# From Brain to Blood: Uncovering Potential Therapeutical Targets and Biomarkers for Huntington’s Disease Using an Integrative RNA-Seq Analytical Platform (BDASeq^®^)

**DOI:** 10.3390/cells14130976

**Published:** 2025-06-25

**Authors:** João Rafael Dias Pinto, Benedito Faustinoni Neto, Luciana Munhoz, Irina Kerkis, Rodrigo Pinheiro Araldi

**Affiliations:** 1BioDecision Analytics Ltd., São Paulo 01451-917, SP, Brazil; joao.dias@biodecesionanalytics.com (J.R.D.P.); bfaustinoni@gmail.com (B.F.N.); 2Post-Graduation Program in Structural and Functional Biology, Paulista School of Medicine, Federal University of São Paulo (UNIFESP), São Paulo 04023-062, SP, Brazil; 3Brazilian Huntington Disease Union House, Scientific Committee, São Paulo 01320-010, SP, Brazil; dra.lucimunhoz@usp.br; 4Genetics Laboratory, Butantan Institute, São Paulo 05503-900, SP, Brazil

**Keywords:** Huntington’s disease, RNA-Seq, propensity score matching, druggable genes, FTH1, biomarker, neuroinflammation, blood–brain barrier, CAG repeat, transcriptomics

## Abstract

Background: Huntington’s Disease (HD) remains without disease-modifying treatments, with existing therapies primarily targeting chorea symptoms and offering limited benefits. This study aims to identify druggable genes and potential biomarkers for HD, focusing on using RNA-Seq analysis to uncover molecular targets and improve clinical trial outcomes. Methods: We reanalyzed transcriptomic data from six independent studies comparing cortex samples of HD patients and healthy controls. The Propensity Score Matching (PSM) algorithm was applied to match cases and controls by age. Differential expression analysis (DEA) coupled with machine learning algorithms were coupled to identify differentially expressed genes (DEGs) and potential biomarkers in HD. Results: Our analysis identified 5834 DEGs, including 394 putative druggable genes involved in processes like neuroinflammation, metal ion dysregulation, and blood–brain barrier dysfunction. These genes’ expression levels correlated with CAG repeat length, disease onset, and progression. We also identified FTH1 as a promising biomarker for HD, with its expression downregulated in the prefrontal cortex and upregulated in peripheral blood in a CAG repeat-dependent manner. Conclusions: This study highlights the potential of FTH1 as both a biomarker and a therapeutic target for HD. Advanced bioinformatics approaches like RNA-Seq and PSM are crucial for uncovering novel targets in HD, paving the way for better therapeutic interventions and improved clinical trial outcomes. Further validation of FTH1′s role is needed to confirm its utility in HD.

## 1. Introduction

Huntington’s Disease (HD, OMIM 143100) is a rare and fatal monogenic autosomal-dominant neurodegenerative disorder characterized by progressive motor dysfunction, neuropsychiatric symptoms, and cognitive decline [1,2].

First described in 1872 by George Huntington as a hereditary movement disorder, with a spectrum of choreiform movements, rigidity, cognitive impairments, and behavioral disturbances in families from Long Island (New York, NY, USA) [3,4], the cause of HD was only resolved in 1993, when The Huntington’s Disease Collaborative Research Group identified the expansion of cytosine–adenine–guanine (CAG) trinucleotide (>36 CAG repeats) in the first exon of the *HTT* gene (*locus* 4p16.3) as HD’s cause [5].

With more than 26,905 published articles since 1903 and 256 clinical trials, with 114 of them evaluating the safety and/or eventual benefits of investigational products in 12,483 enrolled subjects (according to PubMed and Clinical.Trial.gov, respectively—data up to 23 May 2025), the disease remains lacking in treatments capable of modifying its natural history even 153 years since the first report on HD. Approved drugs for HD treatment, tetrabenazine (Xenazine^®^), deutetrabenazine (Austedo^®^), and valbenazina (Ingrezza^®^), only offer benefits for chorea treatment [6,7,8,9]. However, by causing postsynaptic depletion of neuroactive monoamines, the use of these drugs is often accompanied by a range of adverse effects (AEs), such as sedation, drowsiness, fatigue, depression, and anxiety, which can hinder daily functioning and exacerbate the already debilitating symptoms of the disease [6,9,10,11,12,13,14,15].

Given the absence of disease-modifying therapies, identifying new therapeutic targets is imperative. In this context, RNA sequencing (RNA-Seq) emerges as a powerful tool to uncover druggable genes and biomarkers that could guide drug repositioning efforts or foster novel drug development through computer-aided drug design (CADD) [16,17].

The RNA-Seq workflow begins with RNA extraction from biological samples, followed by either mRNA enrichment or rRNA depletion, as protein-coding RNAs are the primary focus for computer-aided drug design (CADD). The enriched mRNA is then reverse-transcribed into complementary DNA (cDNA), and a cDNA library is constructed. This library is sequenced at a depth of 10–100 million reads per sample using high-throughput platforms, typically from Illumina™. The resulting sequences undergo rigorous quality control assessments, evaluating parameters such as read accuracy (Phred score), exogenous RNA contamination (%GC content), and fragment length distribution. After quality control, reads are aligned to a reference genome to determine the set of expressed genes. Subsequently, mapped reads are quantified, normalized, and statistically compared to controls to identify differentially expressed genes (DEGs) [18,19].

However, the identification of DEGs can be influenced by the theoretical assumptions, normalization strategies, and statistical corrections for multiple comparisons (e.g., *p*-value adjustment or false discovery rate, FDR) employed by different differential expression analysis (DEA) methods [20,21,22]. Given the lack of a universally accepted gold standard method for DEA, BioDecision Analytics™ developed the Biomarker Discovery Algorithm (BDASeq^®^) (registered under Brazilian National Institute of Industrial Property number BR512025001097-4) to address these analytical challenges [23,24].

BDASeq^®^ is an integrative analytics-based tool designed to identify putative therapeutic targets (druggable genes) and/or biomarkers of diseases. BDASeq^®^ is divided into three layers: (i) Processing data, which allows the direct uploading of samples from directory or public BioSamples from the Sequence Read Archive (SRA) database using the SRA Toolkit, analyzes sequencing quality control (FastQC [25] and MultiQC [26]), aligns (STAR [27]), and counts the mapped reads (featureCounts [28]), thus generating a counting matrix. (ii) Differential expression, which combines the results of eight different statistical methods that come from the following: (i) Bioinformatics (DESeq2 [22], edgeR [29], limma-Voom [30], NOISeq [31], EBSeq [32], dearseq [33]) and (ii) classical statistics (Wilcoxon [34] and Firth logistic regression [35]). This ensemble of different techniques designed for DEA is performed by the innovative Recursive Method Combination (RMC), which simultaneously controls type I and II errors, reducing analytical bias and generating a gene classification list. Finally, there is the (iii) gene target search, which uses Data Analytics and artificial intelligence (AI) algorithms to identify potential druggable genes and/or biomarkers, as schematically illustrated in Figure 1. In addition, BDASeq^®^ also counts on the optional control sample selection tool. Using the Propensity Score Matching (PSM) algorithm [36,37], a quasi-experimental method introduced by Rosenbaum and Rubin [36], BDASeq^®^ reduces case–control selection bias, increasing target gene accuracy identification for CADD [18,24,38]. Additionally, the use of an unsupervised algorithm devoted to identifying outliers through clustering approaches was employed to remove undesired samples from the expression matrix.

This large-scale transcriptomic analysis, which included the reanalysis of 353 samples from 12 independent BioProjects available in the SRA database, represents the most comprehensive RNA-Seq-based study of HD to date. Supported by the International Huntington Association (IHA), Brazil Huntington Association (ABH), and the Brazilian Huntington Union House (CBUA), our findings uncovered 394 novel putative druggable genes and 18 potential blood biomarkers linked to clinical–pathological features of HD, including FHT1. These discoveries pave the way for new therapeutic strategies and provide crucial tools for monitoring disease progression and therapeutic responses in future clinical trials.

## 2. Materials and Methods

### 2.1. Ethical Approval

This study utilized sequencing data from individuals who tested positive for the HD gene and neurologically normal individuals in the SRA public database. Formal ethical approval was not required, as the identities of all participants remain anonymous according to SRA ethical guidelines. No additional institutional review board (IRB) approval was necessary.

### 2.2. BioProject (Data) Description

There were a total of 778 brain BioSamples (*n* = 778) from 12 independent BioProjects (PRJNA271929, PRJNA316625, PRJNA670925, PRJEB59710, PRJNA253002, PRJNA557205, PRJNA602538, PRJNA531456, PRJNA398545, PRJNA729761, PRJNA388174, and PRJNA755746) containing transcriptomic data (FASTQ files) from the prefrontal (Broadman’s area 9) and motor cortex (Broadman’s area 4); 233 blood BioSamples (n = 233) from three independent BioProjects (PRJNA225101, PRJNA559372, and PRJNA261011) of potential cases and controls were identified in the SRA database as potential data for the analysis, as illustrated in Figure 2. The study considered eligible BioSamples from humans carrying the *mHTT* gene (cases), as confirmed by genetic diagnostics (declared in BioProject metadata, cases), or those not carrying the *mHTT* gene whose cause of death was unrelated to any neurodegenerative disorder (controls). To avoid case–control bias, control samples without age matching with cases were excluded using the MatchIt package in R [39], as implemented into the BDASeq^®^ platform (Propensity Score Matching (PSM) algorithm). Only samples passing basic quality control measures, such as RNA integrity and sequencing depth thresholds, were included for further analysis.

### 2.3. Control Sample Selection by Age Distribution

In a previous study, we showed that the use of older controls indeed leads to the misidentification of DEGs, negatively affecting the discovery of druggable genes and/or biomarkers. This finding highlights the critical importance of considering the age of control samples in RNA-Seq analysis and suggests that age matching should be included as the best practice for such investigations [18]. Therefore, using the MatchIt package in R [39], implemented in the BDASeq^®^ platform, we created a virtual group of pre-eligible BioSamples from neurologically normal individuals, age-matched to HD gene-positive individuals at a 1:1 ratio (HD gene-positive subject: neurologically normal subject). To validate the outcomes of the control sample selection, we visually assessed the age distribution before and after PSM through both boxplot and density plot. Additionally, statistical confirmation of the quality of age distribution was carried out using Student’s *t*-test (*p*-value > 0.05), all executed within the R environment (version 4.4.1, available on https://www.r-project.org/). A *p*-value greater than 0.05 was considered indicative of no significant difference between groups after matching.

### 2.4. Data Processing, Differential Expression Analysis, and Target Discovery Using BDASeq^®^

FASTQ files of eligible BioSamples, including age-matched control samples, were downloaded from the SRA database using SRA Toolkit. Pre-eligible BioSamples were subjected to quality control analysis using FastQC [25] and MultiQC [26]. Samples selected for this study met the following criteria: a Phred score above 30, fragment length of 100–150 bp, and %GC of 50% ± 4, according to the recommendations by Conesa et al. [40].

Eligible BioSamples were then aligned with the human reference genome GRCh38 (release 108) using STAR [27]. Mapped reads were counted with featureCounts [28]. This process resulted in a count matrix, which serves as input for the next step differential expression analysis (DEA). Due to the combination of different studies, with different strategies, the batch effects from the raw count matrix were removed by using the CombatSeq algorithm [41]. Bioinformatic tools used for data downloading and processing are implemented in the first layer of BDASeq^®^ (data processing). Outliers samples were filtered out using the combination of unsupervised algorithm with clustering analysis. DEGs analysis was performed using eight different methods: DESeq2 [22], edgeR [29], limma-Voom [30], NOISeq [31], EBSeq [32], dearseq [33], Wilcoxon [34], and Firth logistic regression [35]. Results from these methods are integrated using the Recursive Method Combination (RMC) algorithm to minimize analytical bias. Genes were then classified as zero (ZCGs), low (LCGs), equally expressed (EEGs), non-relevant (NRGs), up- (UGRs) or downregulated genes (DRGs), as previously proposed by us [18]. The criteria of classification used are summarized in Table 1. Bioinformatic and statistics tools used for DEA and the RMC algorithm are part of the second layer of BDASeq^®^ (differential expression). Subsequently, the most relevant genes (UGRs and DRGs) and their association with clinical features of Huntington’s Disease (HD)—including CAG repeat number, age at onset, Vonsattel grade (indicative of brain degeneration), and lifespan—were identified using supervised classification algorithms coupled with feature selection techniques [41]. This feature selection step, combined with feature (i.e., gene) importance measurements, were integrated into the third layer of BDASeq^®^ (Gene Target Search). Results were summarized in tables, heatmaps, and boxplots based on normalized read counts.

## 3. Results

### 3.1. Study Design and Sample Selection

From the 778 BioSamples (*n* = 778) across 12 independent BioProjects (Appendix A) containing transcriptomic data from the prefrontal cortex of HD gene-positive (cases) or neurologically normal individuals (controls), 425 BioSamples (*n* = 425) were excluded for not meeting eligibility criteria. Specifically, the following criteria were used for exclusion: (i) non-human samples (*n* = 50), (ii) non-HD samples (*n* = 226), (iii) samples from non-prefrontal cortex tissue (*n* = 71), (iv) samples failing quality control (*n* = 55), (v) juvenile control samples (*n* = 20), and (vi) asymptomatic FD cases (*n* = 3). After this exclusion, 353 BioSamples (*n* = 353) were deemed eligible, comprising 146 gene-positive cases and 207 neurologically normal controls (Figure A1).

To minimize case–control bias, the BDASeq^®^ tool employed a PMS algorithm to select the age-matched subset of control samples (age-matched controls, *n* = 146, 59.1 ± 12.3 years) comparable to the HD group (*n* = 146, 57.9 ± 14.4 years) Consequently, 61 unmatched control samples (80.8 ± 14.4 years) were excluded (Figure 3). In total, 296 prefrontal cortex samples were included in the final analysis: 146 from HD gene-positive cases and 146 matched controls. To our knowledge, this constitutes the largest transcriptomic study of HD compiled into data, as recognized during the MENA Congress for Rare Disease 2024 [24,38] and by the FAPESP Agency (https://agencia.fapesp.br/a-startup-supported-by-fapesp-is-conducting-one-of-the-largest-studies-on-huntingtons-disease/51647, accessed on 23 June 2025).

### 3.2. Aging Promotes Transcriptional Alterations in the Prefrontal Cortex

As anticipated, a statistically significant difference was observed between HD cases and controls (Figure A3). Reanalysis of one of the 12 BioProjects (PRJNA271929) revealed that using control samples without proper age matching with cases increases both type I and II errors, leading to misidentification of DEGs [18].

Dimensionality reduction analysis using Uniform Manifold Approximation and Projection (UMAP) of the transcriptomic profiles from the 353 BioSamples analyzed in this study showed that cases and controls were distinct in the multidimensional space (Figure A3A), confirming that HD induces transcriptional changes in the prefrontal and motor cortex, as expected. However, clustering analysis revealed that 353 BioSamples are grouped in three clusters (Figure A3B): C0 and C1, predominantly composed of neurologically normal controls, and C2, primarily comprising HD gene-positive individuals (Figure A3C). Further analysis of the control groups revealed that cluster C0 consisted of neurologically normal controls who were older than those in cluster C1, as demonstrated in Figure A3D. These findings confirm that aging contributes to transcriptional changes in the prefrontal cortex and emphasizes the importance of carefully selecting control samples to prevent target misidentifications, as we previously reported [18]. Using clustering techniques, it was possible to identify and remove 20 samples considered as outliers capable of drastically affecting the differential expression analysis (depicted as C-1 cluster in Figure A3B). The exclusion of these 20 samples were based on the combination of 2D UMAP representation of the bulk gene expression (removing zero- and low-count genes) with DBSCAN analysis. DBSCAN, in nature, works by defining clusters as densely packed regions separated by regions of lower density. This approach allows DBSCAN to discover and label sparse distributed samples as noise, i.e., outliers.

### 3.3. BDASeq^®^ Identified 394 Potential Druggable Targets for HD

Transcriptomes from the cases and controls were compared by using the BDASeq^®^. Of the 62,703 transcripts annotated in the human reference genome, 4005 transcripts were classified as ZCGs, and 32,533 were classified as LCGs—both categories corresponding genes that are not expressed by the prefrontal and motor cortex. Among the expressed genes (normalized counts > 10), 12,901 transcripts were categorized as EEGs, and thus were excluded as potential therapeutic targets for HD. A total of 7430 transcripts were identified as NRGs, being statistically significant by adjusted *p*-value (FDR) but lacking a meaningful log2FC, and were therefore considered suboptimal for drug development (criteria of NRG definition available on Table 1). Conversely, BDASeq^®^ identified 5834 DEGs, comprising 3729 as URG and 2105 as DRG transcripts (Figure 4, Appendix A).

Of these 5834 DEGs, 2896 encode proteins—representing 49.64% of total the DEGs—and were considered potential druggable targets (Appendix A). Among these protein-coding DEGs, 83.53% (2057/2377) were concordantly identified by both BDASeq^®^ and the traditional DESeq2 pipeline (Figure 5A, Appendix A). However, BDASeq^®^ uniquely identified 839 additional protein-coding targets not detected by DESeq2—putative false negative DEGs—while DESeq2 identified 320 targets not captured by BDASeq^®^—putative false positive (Figure 5A, Appendix A). These discrepancies are consistent with expectations as the integration of the PSM and RMC within BDASeq^®^ is designed to simultaneously control type I and type II errors, thereby refining gene classification, as illustrated in the Sankey diagram (Figure 5B).

Traditionally, DEGs are ranked solely by their log2FC, optionally combined with FRD-adjusted *p*-values (FDR), to shortlist the most significantly up- or downregulated genes for further consideration as drug targets or biomarkers [42]. However, as previously discussed [18], this approach can result in target misclassification. To address this limitation, BDASeq^®^ incorporates an AI-driven feature selection algorithm that prioritizes candidate genes based on their expression distribution between cases and controls rather than fold-change magnitude alone. Using this algorithm, BDASeq^®^ counts on a feature selection AI-based algorithm that appropriately selects target genes based on their distribution between cases and controls. Using this algorithm, BDASeq^®^ identified 394 protein-coding genes as druggable targets and/or biomarkers for HD (Figure 6, Appendix A).

Comparative analysis of normalized expression values between HD and control samples showed that 210 of these genes were upregulated, and 184 were downregulated in the prefrontal cortex and motor cortex of HD gene-positive individuals (Figure 7, Appendix A). Furthermore, the Gini coefficient—a measure of statistical dispersion reflecting intergroup expression variability—ranged from 0.3 (*ANKRD62*) to 1.0 (*PITX1*), supporting the robustness of the gene selection process (Appendix A).

Notably, when comparing BDASeq^®^ results to those from DESeq2, only 103 of the 394 genes (26.14%) were identified by both approaches (Figure A4). This indicates that BDASeq^®^’s AI algorithm revealed 291 novel targets that would have been overlooked using conventional log2FC-based ranking (Figure A4), highlighting its potential to uncover new avenues for drug development, repositioning, and prognostic biomarker discovery in HD.

### 3.4. Putative Druggable Genes Identified by BDASeq^®^ Are Associated with Dysregulated Biological Processes in HD

Given that a druggable gene is ideally defined as a protein-coding gene linked to a disease-specific biological process that can be therapeutically modulated by a drug [43], the 394 protein-coding genes identified by BDASeq^®^ were subjected to functional enrichment analysis using multiple Gene Ontology (GO) databases. The enrichment results revealed that these candidate genes are significantly involved in biological processes and molecular functions known to be disrupted in HD. These include responses to metal ions, cholesterol metabolism, neuroinflammation, transport across the blood–brain barrier, negative regulation of the cell cycle, disruption of membrane scaffolding, axon guidance, and neurotransmission (Figure 8, Table 2, Appendix A). These findings reinforce the biological relevance of the BDASeq^®^-identified targets, as they are functionally tied to core pathological mechanisms of HD and may thus serve as promising candidates for therapeutic development or repositioning.

### 3.5. Target Genes Identified by BDASeq Are Involved in Aging-Related Processes Exacerbated in HD

Supporting previous evidence that HD exacerbates aging-related biological processes [67,68,69]; analysis revealed that 149 out of the 394 BDASeq-identified target genes (37.81%) are associated with aging signatures across multiple tissues, including the brain (Figure A5, Appendix A). These findings provide a molecular basis for the systemic clinical manifestations of HD beyond neurodegeneration. In addition to brain dysfunction, HD is known to cause: (i) skeletal muscle impairments, associated with chloride and potassium channel dysfunction, resulting in muscle hyperexcitability, hypertonicity, twitching, and atrophy [70,71,72,73]; (ii) ocular alterations, including retinal nerve fiber layer thinning [74,75,76,77,78]; (iii) bone marrow disruption; (iv) immune system dysregulation, leading to platelet abnormalities [76] and increased production of pro-inflammatory cytokines [79]. Altogether, the 149 aging-associated target genes identified by BDASeq^®^ offer promising opportunities for the development of therapeutic strategies that address both the neurological and systemic features of HD.

### 3.6. Target Genes Identified by BDASeq Are Deregulated in Brain Regions Affected by HD

Analysis revealed that 391 out of the 394 BDASeq-identified target genes (99.23%) are expressed in brain regions known to be progressively affected in HD, including the cingulate cortex, striatum, caudate nucleus, thalamus, motor cortex, substantia nigra, and cerebellum (Figure A6, Appendix A). As expected, the majority of these 391 are expressed in brain regions most vulnerable to HD pathology, such as the striatum [70,71], cingulate [72,73], and prefrontal cortex [74,75,76] (Figure A6, Appendix A). This observation aligns with the known neuroanatomical progression of HD, as the prefrontal cortex maintains strong functional connectivity with both the cingulate cortex [77] and striatum [78], which may explain the overlapping gene expression profiles observed. These findings reinforce the relevance of the identified targets, as they are localized to key regions implicated in HD pathophysiology and may contribute to region-specific neurodegeneration.

### 3.7. BDASeq^®^ Identified CAG- and Onset-Age-Related Target Genes

Statistical analysis revealed a strong and significant negative correlation between the CAG repeat length and onset age (*p*-value = 0.0000 and *R* = 0.8160, Figure 9A) as well as between CAG repeat length and death age (*p*-value = 0.0000 and *R* = 0.8538, Figure 9B). Conversely, a significant positive correlation was observed between age at onset and age at death (*p*-value = 0.0000 and *R* = 0.9132, Figure 9C).

Given that the CAG repeat length is critical clinical–pathological hallmark of HD closely associated with both disease onset and progression, we applied the BDASeq^®^ AI algorithm to further interrogate the 394 previously identified protein-coding genes (considered potential druggable targets and/or biomarkers for HD). This analysis identified five DRGs and seven URGs whose expression levels are significantly associated with CAG repeat length, highlighting their potential as CAG-related target genes (Table 3, Figure 10). Functional enrichment analysis of these genes revealed relevant biological processes, notably involving DHDDS, a gene implicated in glycosylation and neurodegenerative disease mechanisms.

Moreover, considering the tight relationship between CAG repeat length and disease onset (Figure 9A), and the positive correlation between onset and age at death (Figure 9B,C), the BDASeq^®^ AI algorithm was also used to identify onset age–related target genes. This analysis yielded 59 potential druggable genes and/or biomarkers, including 33 DRGs and 26 URGs whose expression levels significantly correlate with age at onset (Table 4, Figure 11).

Together, these findings underscore the potential of BDASeq^®^ in identifying clinically relevant gene targets that are not only associated with HD pathology but also with key disease-modifying factors such as CAG repeat length and age of symptom onset.

### 3.8. BDASeq^®^ Identified Target Genes Associated with Brain Degeneration

Analysis of the Vonsattel grade data (available on the metadata of the analyzed BioSamples), revealed that earlier onset age (Figure 12A) and earlier age at death were associated with more severe brain degeneration (Figure 12B). Additionally, increased brain degeneration was correlated with longer CAG repeat (Figure 12C). reinforcing the strong relationship between CAG repeat length and neurodegenerative progression in HD.

Although BDASeq^®^ identified 12 genes associated with CAG repeat length (Table 5, Figure 13) that could potentially serve as biomarkers, their therapeutic value may be limited. This limitation arises because, despite pharmacological modulation, the persistent expression of *mHTT* may continue to dysregulate these targets, reducing the efficacy of potential therapies due to competitive interference. To address this challenge, the BDASeq^®^ AI algorithm also prioritized gene targets based on their association with Vonsattel grade—a direct indicator of neurodegeneration and indirectly linked to CAG repeat length (Figure 12C). This approach led to the novel identification of 84 brain degeneration-related genes, comprising 47 DRGs and 37 URGs, which may serve as viable therapeutic targets to counter HD-related neurodegeneration (Table 5, Figure 13).

Furthermore, recognizing that both brain degeneration and CAG repeat length correlate with age of death (Figure 12B), this study calculated life expectancy in HD gene-positive individuals as the difference between age of death and age of symptom onset. Using this metric, BDASeq^®^ identified 16 genes—including 13 DRGs and 3 URGs—whose expression is strongly associated with patient lifespan (Table 6, Figure 14). These genes represent promising targets for therapeutic strategies aimed at altering the disease course and extending life expectancy in HD.

### 3.9. BDASeq^®^ Identified for First-Time Blood Biomarkers for HD, Including FTH1

In the analysis of three additional BioProjects, PRJNA225101 (*n* = 124 BioSamples), PRJNA559372 (*n* = 93 BioSamples), and PRJNA261011 (*n* = 16 BioSamples)—which included blood RNA-Seq datasets from HD gene-positive individuals and controls–BDASeq^®^ identified only 20 DEGs in the blood of HD patients (Table 7). However, two of these DEGs are non-coding genes (ENSG00000250568 and ENSG00000260035, Table 7), reducing the number of relevant DEGs to 18 genes. These genes were removed from downstream analysis, once for drug design or diagnosis, protein-coding genes are the primary targets of many drugs and antibodies.

Among these 18 DEGs, only one gene, *FTH1,* was also identified in the prefrontal and motor cortex of HD in gene-positive individuals (Figure 15A). Interestingly, *FTH1* was found downregulated in a CAG repeat length-dependent manner (Figure 15B) and upregulated in the peripheral blood of HD gene-positive individuals (Figure 15C). As the major subunit of ferritin, FTH1 is responsible for intracellular iron storage and protection against iron-induced oxidative stress. This finding positions FTH1 as a promising candidate for biomarker for HD.

Furthermore, ARRB2 and PFKFB3 are promising biomarker candidates for Huntington’s disease (HD) due to their involvement in key pathological processes—*ARRB2* in neuronal signaling and cell death, and *PFKFB3* in glycolytic metabolism and energy stress. Both genes are expressed in the brain, including the prefrontal cortex, and in peripheral blood, which supports their potential for minimally invasive monitoring. Although CAG repeat expansion does not directly regulate *ARRB2* or *PFKFB3*, it induces cellular conditions such as stress, metabolic shifts, and synaptic dysfunction, which in turn affect these genes. Consequently, their altered expression and functional activity may serve as indirect indicators of CAG-induced pathology, reinforcing their potential as biomarkers for HD. Changes in their expression could reflect disease progression and therapeutic response. However, further validation is needed to confirm their specificity and sensitivity within the context of HD.

## 4. Discussion

HD was first reported over 150 years ago, yet it remains without drugs capable of altering its natural history. Current approved treatments, such as tetrabenazine, deutetrabenazine, and valbenazina, provide limited therapeutic benefits, mainly addressing chorea in the early stages of the disease. However, these treatments are associated with multiple adverse effects, including depression, which increases the risk of suicide. The absence of disease-modifying drugs that provide wider therapeutic benefits or postpone neurodegeneration is largely due to the complexity of the disease, the lack of suitable animal models, and the limited number of druggable genes that can be explored through computer-aided drug design (CADD) [16,17]. In this context, RNA-Seq analysis combined with different analytical approaches are a promising tool for identifying potential druggable genes.

To uncover such genes, at least six independent studies have comparatively analyzed the transcriptomes of prefrontal cortex postmortem samples from symptomatic individuals carrying the *mHTT* gene (cases) and neurologically normal controls [75,76,79,80]. These studies focused on prefrontal cortex and caudate nucleus, since about 90% of striatal neurons, primarily affected by the disease, are lost in late-stage HD, making it difficult to study striatal postmortem samples from individuals with HD due to the scarcity of neurons in highly degenerated tissue, as previously discussed by us [18]. However, studies based on structural magnetic resonance imaging (MRI) evidenced that, in late-stage HD, BA9 exhibits loss of projection neurons in layers III, V, and VI and glial density increases in deeper layer (VI) consistent with cortical degeneration [81,82]. These results make BA9 an important brain area to be explored to identify possible pharmacological/prognostic targets for HD [18].

Despite these substantial efforts, the DEGs identified by these studies have not led to the discovery of useful druggable genes or blood biomarkers for HD [75,76,79,80,83,84]. A common theme across these studies is the identification of genes belonging to the heat shock family or genes involved in inflammatory processes as potential therapeutic targets and/or biomarkers [75,76,79,80,83,84].

Reanalyzing the transcriptomic dataset of one of these studies [75], we demonstrated that failure to match cases and controls by age negatively affects DEG identification, leading to an increase in both false positive and false negative errors [18]. Using a PSM algorithm to create an age-matched control group, we were able to identify multiple putative druggable genes involved in HD pathophysiology that do not belong to the heat shock family [18]. Further examination of previously published transcriptomic studies on HD [75,76,79,80,83,84] revealed that differences in bioinformatic pipelines, including various differential expression analysis (DEA) methods, different normalization approaches, and the small sample sizes, might contribute to the lack of consensus regarding DEGs. To address this limitation, we aggregated RNA-Seq data from multiple studies available in the SRA database, which significantly increased the sample size. This strategy was crucial for DEG identification, given that HD is a heterogeneous disease. Through this approach, we identified 353 pre-eligible BioSamples. Using the PSM function in BDASeq^®^, we obtained 292 eligible BioSamples (146 age-matched controls, and 146 cases) for further analysis, making this the largest transcriptomic study on HD compiled into data, as recognized by the International Huntington Association (IHA), Brazil Huntington Association (ABH), and *Fundação de Amparo à Pesquisa do Estado de São Paulo* (FAPESP, available on https://agencia.fapesp.br/a-startup-supported-by-fapesp-is-conducting-one-of-the-largest-studies-on-huntingtons-disease/51647, accessed on 23 June 2025).

Applying the RMC algorithm, BDASeq^®^ identified 5834 DEGs, including 394 putative druggable genes in the prefrontal and motor cortex of HD-positive individuals. Comparative analysis of the expression levels (read counts) of these 394 genes confirmed their differential expression in the prefrontal cortex of HD patients. Functional enrichment analysis revealed that these 394 putative therapeutic targets are involved in key HD-related pathophysiological processes, including metal ion dysregulation, cholesterol metabolism disruption, neuroinflammation, impaired blood–brain barrier transport, neurotransmission dysfunction, and axon guidance disturbances. These results suggest that BDASeq^®^ effectively identified candidate genes for therapeutic targeting in HD.

In addition, BDASeq^®^ also highlighted potential therapeutic target genes whose expression levels are associated with CAG repeat length, onset age, Vonsattel grade (or brain degeneration), and lifespan. Although further validation is required to confirm their role in these processes, these genes hold promise for drug design or repositioning to provide broader therapeutic benefits than those offered by current drugs like tetrabenazine, deutetrabenazine, or valbenazine, as well as investigational compounds like pridopidine, which are focused solely on chorea.

In the context of drug development, a significant challenge, particularly in clinical trials, is demonstrating the efficacy of investigational products. This challenge is compounded by the lack of reliable biomarkers, which restrict efficacy assessments to subjective measures like the Unified Huntington’s Disease Rating Scale (UHDRS). While the UHDRS’ Total Motor Score (TMS) and Total Functional Capacity (TFC) assessments are widely used and have acceptable interrater reliability, they ultimately depend on the clinical experience of the rater and lack precision in certain contexts. Therefore, the discovery of HD biomarkers is crucial for improving clinical trial outcomes.

Ideally, a biomarker should be a biomolecule whose expression is differentially regulated under pathological conditions and can be easily monitored in accessible tissues, preferably peripheral blood [85].

Several studies have aimed to identify HD biomarkers in the peripheral blood of HD gene-positive individuals [83,84,86]. However, these studies have not yielded useful biomarkers, as pointed out by Mitchell et al. [85] in the TRACK-HD study. This failure can be attributed to a lack of advanced analytical technologies that combine AI and bioinformatics to integrate transcriptomic data with clinical–pathological features. In our study, we also analyzed three additional BioProjects containing transcriptomic data from the blood of HD gene-positive individuals. BDASeq^®^ identified 18 potential blood biomarkers. Interestingly, only one of these biomarkers (*FTH1*) was found among the 394 potential therapeutic targets differentially expressed in the prefrontal cortex of HD individuals.

Recent studies highlight *PFKFB3* and *ARRB2* as potential biomarkers and therapeutic targets in HD. *PFKFB3*, a regulator of glycolytic metabolism, is implicated in neurodegenerative diseases like Alzheimer’s, where it affects amyloid-beta accumulation and neuronal dysfunction [87]. In HD, its expression may reflect metabolic shifts and neuronal stress. Similarly, *ARRB2* (Beta-Arrestin 2) is involved in neurodegeneration by interacting with pathways linked to protein accumulation, such as amyloid-beta in Alzheimer’s and tau in frontotemporal dementia. *ARRB2* also influences neuroinflammation, synaptic dysfunction, and may regulate alpha-synuclein in Parkinson’s disease, suggesting a similar role in HD [88]. Additionally, *ARRB2* mediates dopaminergic signaling in the hippocampus, which is relevant to HD pathology [89]. Together, these genes contribute to key mechanisms in HD, including cellular stress, metabolic dysfunction, and protein aggregation, supporting their potential as biomarkers and therapeutic targets. However, further research is needed to validate their role in HD.

Given that an ideal biomarker should be differentially expressed in both the primary tissue affected by the disease but also be differentially expressed in peripheral blood, with its expression linked to disease phenotype or pathophysiological processes [85], FTH1 emerges as a promising biomarker for HD. The FTH1 gene was found in the prefrontal cortex in a CAG repeat length-dependent manner but upregulated in the peripheral blood of HD gene-positive individuals.

The *FTH1* gene encodes the ferritin have chain 1 (FTH1), a critical protein involved in iron homeostasis [90]. FTH1 acts as the primary cytosolic iron storage protein, buffering against oxidative stress [91]. Elevated levels of FTH1 helps maintain iron balance and mitochondrial homeostasis, protecting cells, including neurons, from oxidative stress [91]. FTH1 is also involved in innate immune system pathways by regulating iron levels and influencing inflammation [91]. In addition, FTH1 is involved in vesicle transport in neurons, particularly in iron storage and neuron survival [92].

Exosomes secreted by oligodendrocytes, which carry FTH1, provide antioxidant protection to neurons against iron-induced cytotoxicity [92]. Interestingly, recent studies have shown that exosomes play a role in iron transport across the blood–brain barrier [93], which may explain the compensatory upregulation of FTH1 observed in the blood of HD-positive individuals.

In HD, FTH1 is associated with ferroptosis, a form of iron-dependent cell death driven by lipid peroxidation, as revised by Jamal et al. [94]. Interesting, downregulation of FTH1 has also been reported as a key mediator of ferroptosis and oxidative stress in other neurodegenerative diseases, such as Parkinson’s (PD) and Alzheimer’s disease (AD) [94,95,96,97]. Consistent with these findings, Li et al. [98] showed that miR-335 promotes ferroptosis in vitro and in vivo models for PD by targeting *FTH1.*

Although serum ferritin levels have been investigated in HD for over three decades [99,100], these studies typically measure total ferritin protein, which may explain the discrepancies observed, with some studies reporting decreased ferritin levels [99,100], while our study shows *FTH1* upregulation in peripheral blood. This highlights the potential of FTH1 as a novel biomarker for HD. However, further studies using qRT-PCR to assess *FTH1* expression and immunoassays to measure FTH1 levels are needed to validate its utility as a biomarker for HD.

## 5. Conclusions

In conclusion, this study demonstrates the effectiveness of different analytical strategies and methods, consisting of the BDASeq^®^, in identifying potential therapeutic targets and biomarkers for HD through a comprehensive transcriptomic analysis. The application of PSM and aggregation of RNA-Seq data from multiple studies allowed for a robust identification of 394 druggable genes, which are implicated in key HD-related processes such as neuroinflammation, metal ion dysregulation, and impaired blood–brain barrier transport. Among these, FTH1 stands out as a promising biomarker due to its differential expression in both the prefrontal cortex and peripheral blood of HD gene-positive individuals. The upregulation of FTH1 in peripheral blood, combined with its role in iron homeostasis and neuroprotection, underscores its potential as a novel biomarker for HD. This finding is further supported by its involvement in ferroptosis, a form of cell death linked to oxidative stress in neurodegenerative diseases. Despite these promising results, further validation using techniques like qRT-PCR and immunoassays is needed to establish FTH1′s clinical utility as a biomarker for HD. Ultimately, the identification of these genes offers valuable insights into the development of disease-modifying treatments and biomarkers that could enhance clinical trial outcomes and improve patient care in HD.

## 6. Patents

BDASeq^®^ is a computer program developed by BioDecision Analytics^TM^. (João Rafael Dias Pinto, Rodrigo Pinheiro Araldi, and Benedito Faustinoni Neto) and registered in the Brazilian National Institute of Industrial Property (INPI, BR512025001097-4).

## Figures and Tables

**Figure 1 cells-14-00976-f001:**
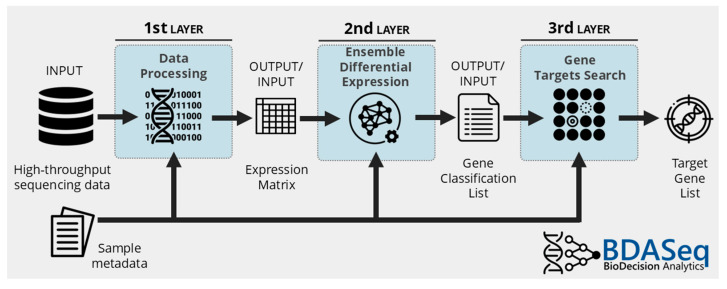
Schematic model of the BDASeq^®^ tool. High-throughput sequencing files/data (FASTQ) are imported into the tool using the SRA Toolkit. To minimize case–control bias, control samples are selected using the Propensity Score Matching (PSM) algorithm. Pre-selected BioSamples from controls and cases are then subjected to quality control analysis using FastQC. Samples that pass quality control are aligned to a reference genome using the STAR aligner and mapped reads are quantified using the featureCounts, generating a count matrix that serves as the input to the next step—differential expression analysis. In the differential expression analysis phase, eight different methods (DESeq2, edgeR, limma-Voom, NOISeq, EBSeq, dearseq, Wilcoxon, and Firth logistic regression) are applied individually. Results are then combined using the disruptive Recursive Method Combination (RMC) algorithm, which simultaneously reduces type I and type II errors, thus minimizing analytical bias. The process also filters out undesired samples through outlier-detection techniques. Following this integrative analysis, genes are classified into six categories: zero-count genes (ZCGs), low-count genes (LCGs), equally expressed genes (EEGs), non-relevant genes (NRGs), up- (URGs) or downregulated genes (DRGs). Finally, in the target gene search layer, artificial intelligence (AI) algorithms are employed to both identify putative target as well as retrieve URGs and DRGs associated with clinical–pathological features of interest (based on the sample metadata variables). BDASeq^®^ is proprietary intellectual property of BioDecision Analytics Ltd., registered in Brazilian National Institute of Industrial Property (INPI, Rio de Janeiro, Brazil, BR512025001097-4).

**Figure 2 cells-14-00976-f002:**
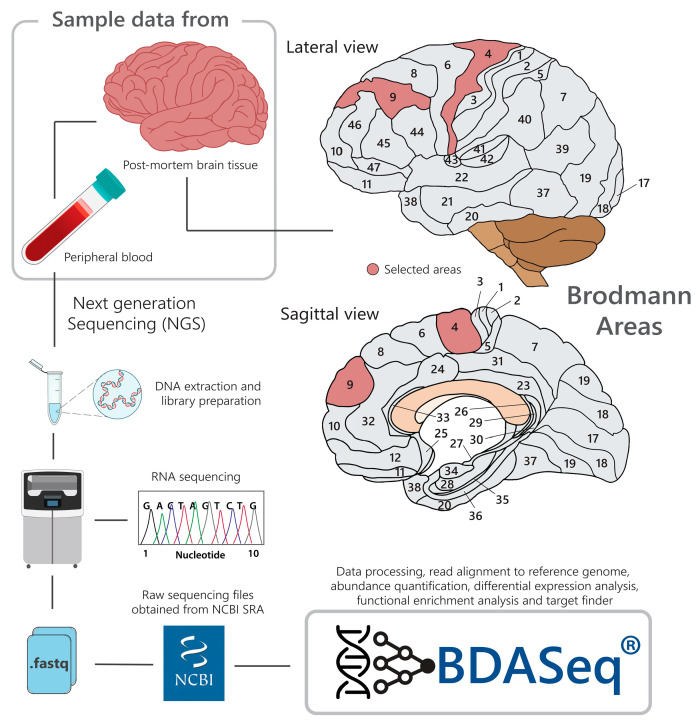
Schematic illustration of the study design. Eligible brain (Broadmann areas (BAs) 4 and 9) and blood BioSamples (FASTQ files) from HD-positive (cases) and neurologically normal controls were downloaded from the SRA database (implemented into BDASeq^®^). Using BDASeq^®^, FASTQ files are processed, being submitted to quality control analysis, sequence alignment/mapping, and read counts. Next, read counts are subjected to differential expression analysis using the RMC algorithm. Differentially expressed genes are combined with clinical–pathological features to identify potential therapeutic targets or biomarkers using AI.

**Figure 3 cells-14-00976-f003:**
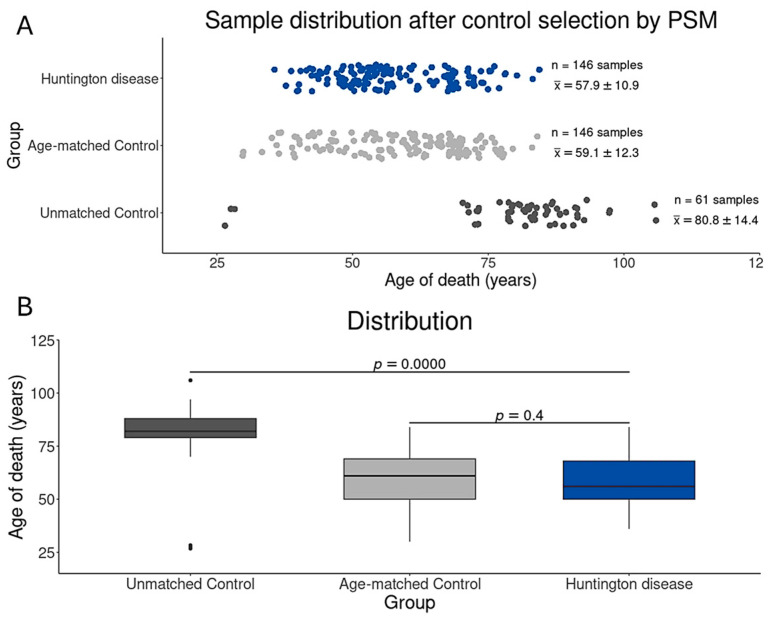
Age distribution after PSM-based sample selection (**A**). Note that the PSM appropriately selects 146 control samples with age distribution (age-matched controls, 59.1 ± 12.3 years) statistically similar to cases (Huntington disease, 57.9 ± 10.9 years) (**B**), excluding 61 unmatched control samples from older neurologically normal individuals (80.8 ± 14.4 years). Statistical analysis was performed using one-way ANOVA, followed by the Tukey post hoc test, both with significance levels of 5%.

**Figure 4 cells-14-00976-f004:**
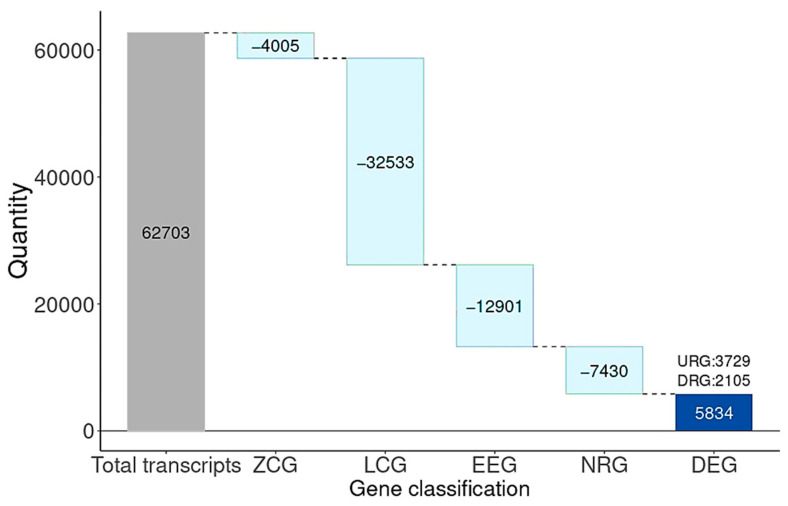
Waterfall plot showing the total number of differentially expressed genes identified in the prefrontal and motor cortex of HD gene-positive individuals in relation to the controls. Note that, from the 62,703 transcripts encoded by the human reference genome, 36,538 (4005 ZCGs and 32,533 LCGs) transcripts are not expressed by the selected brain areas. Fron the expressed genes, 12,901 are equally expressed by cases and controls (EEGs); 7430 transcripts show a statistically significant adjusted *p*-value (FDR), but with no significant log2FC (NGR); and 5834 transcripts are differentially expressed between cases and control. Of these, 3729 are upregulated (URG), and 2105 are downregulated (DRG).

**Figure 5 cells-14-00976-f005:**
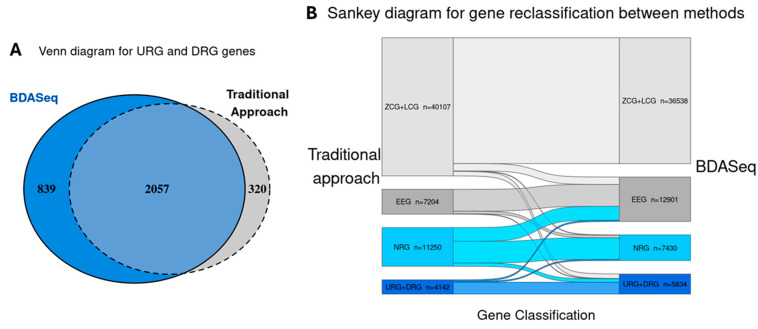
Comparative analysis between BDASeq^®^ and the traditional approach using DESeq2 and the commonly used log2FC criteria for target selection. Venn diagram comparing the protein-coding genes (potential drug targets and/or biomarkers) identified by BDASeq and the traditional approach. Results show that BDASeq^®^ identified 2057 URGs and DRGs in common with the traditional approach but identified 839 potential targets not identified by the traditional approach. Besides this, BDASeq^®^ excludes 320 potential targets identified by the traditional approach (putative false positive targets) (**A**). The Sankey diagram shows that by appropriately selecting age-matched control samples with cases and combining eight different DEA techniques using the innovative RMC algorithm, BDASeq^®^ reclassifies the gene status of the traditional approach (**B**).

**Figure 6 cells-14-00976-f006:**
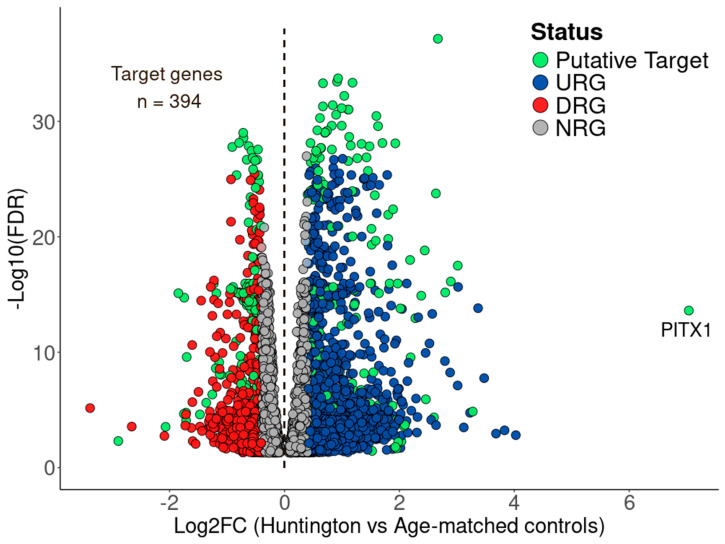
Volcano plot showing the downregulated (DRG, in red) and upregulated (URG, in blue) identified by the RMC algorithm, as well as the putative target protein-coding genes (394) identified in prefrontal and motor cortex of HD-positive individuals using the AI-based feature selection algorithm from these DRGs and URGs.

**Figure 7 cells-14-00976-f007:**
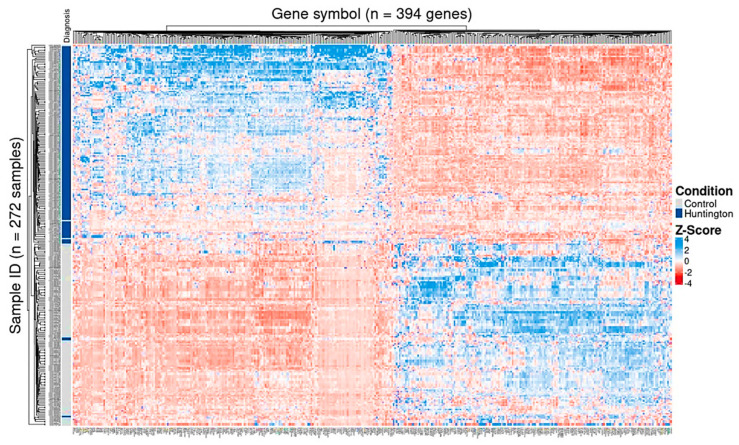
Heatmap showing the normalized counts from the 394 protein-coding genes potentially identified as targets by BDASeq^®^. Results confirm that these genes are differentially expressed in prefrontal and motor cortex from HD-positive individuals when compared to the prefrontal cortex of neurologically normal controls.

**Figure 8 cells-14-00976-f008:**
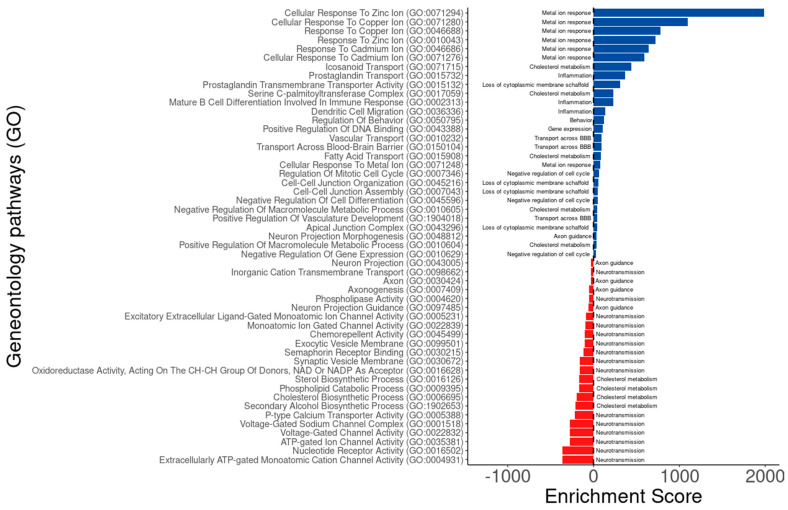
Functional enrichment analysis per overrepresentation (ORA) of the 394 target genes identified by BDASeq^®^. Results show that the target genes positive (in blue) and negative (in red) regulate biological pathways involved in HD pathophysiology, being related to metal ion response, inflammation, behavior, negative regulation of the cell cycle, transport across the brain–blood barrier (BBB), loss of membrane scaffold, axon guidance, neurotransmission, and cholesterol metabolism. Results obtained in terms of Gene Ontology (GO), using the Erichr webservice.

**Figure 9 cells-14-00976-f009:**
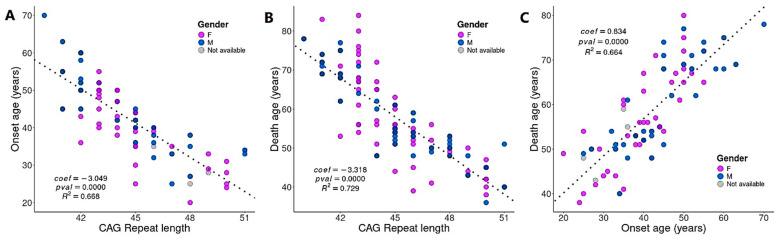
Relevant clinical–pathological features identified using the BDASeq^®^. Results show a negative correlation between the CAG repeat length and the onset (**A**) and death age (**B**), as well as a positive correlation between the onset and death age (**C**). Doted line describe linear regression.

**Figure 10 cells-14-00976-f010:**
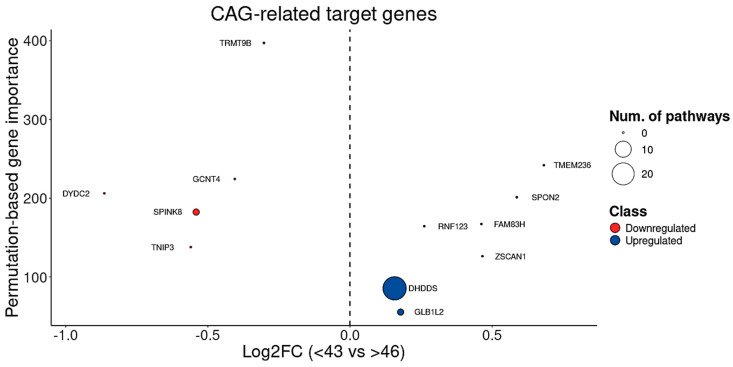
Analysis of the 12 CAG-related target genes showing that, from these genes, five are downregulated and seven are upregulated. Circle diameter indicates the number of pathways in which each gene is involved. The comparison is made considering both superior and inferior extremes of CAG distribution across HD-positive cases.

**Figure 11 cells-14-00976-f011:**
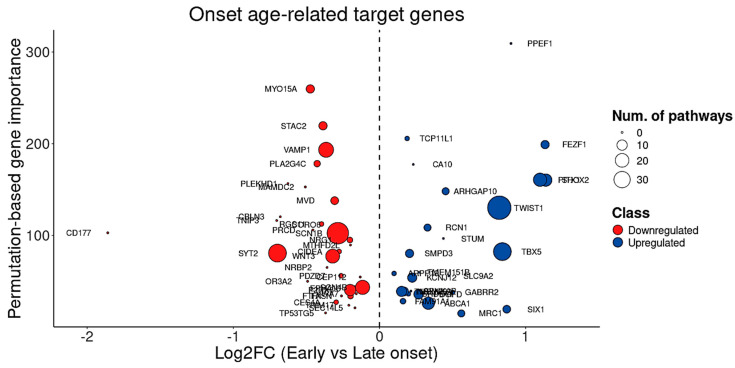
Analysis of the 59 onset age-related target genes showing that, from these genes, 33 are downregulated and 26 are upregulated. Cycle diameter indicates the number of pathways in which each gene is involved. Early onset considered onset age < 35 years whereas late onset considered onset age > 50 years.

**Figure 12 cells-14-00976-f012:**
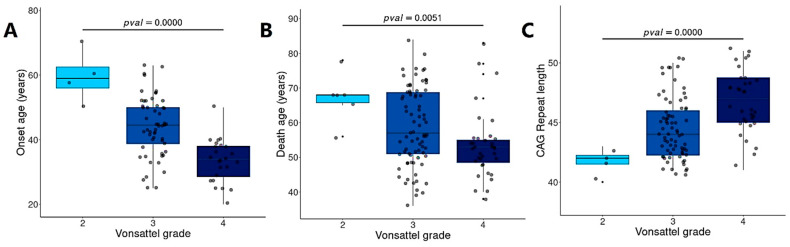
Analysis of brain degeneration based on the Vonsattel grade. Results show that the higher the Vonsattel grade and, therefore, the brain degeneration, the shorter the onset (**A**) and death age (**B**), but the higher the CAG repeat length (**C**). Results were obtained using BDASeq^®^. Statistical analysis was performed using ANOVA, with a significant level of 5%.

**Figure 13 cells-14-00976-f013:**
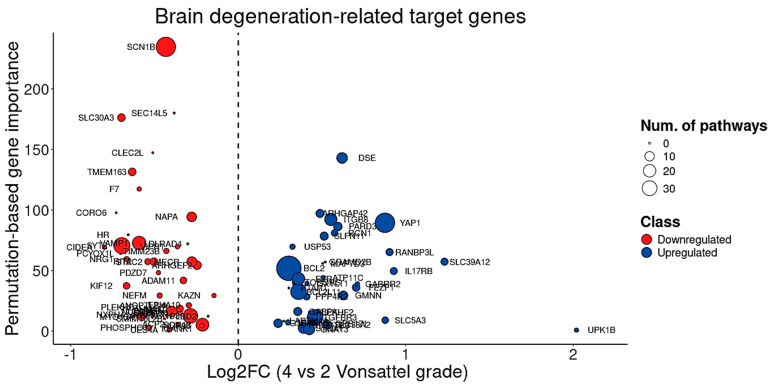
Analysis of the 84 brain degeneration-related target genes showing that, from these genes, 47 are downregulated and 37 are upregulated. Circle diameter indicates the number of pathways in which each gene is involved. In this comparison we used both lower level of degeneration vs. higher level of degeneration (Vonsattel scale 4 vs. 2).

**Figure 14 cells-14-00976-f014:**
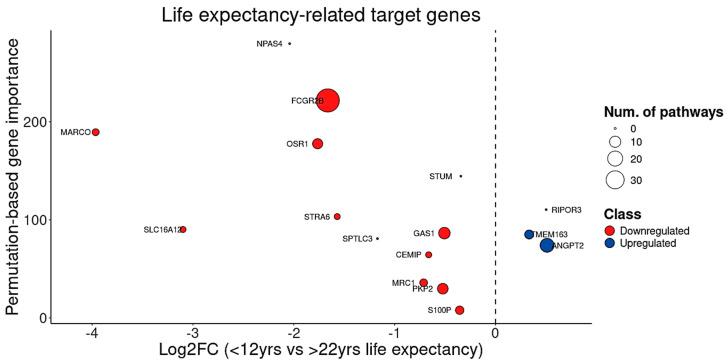
Analysis of the 16 life expectancy-related target genes showing that, from these genes, 13 are downregulated and 3 are upregulated. Circle diameter indicates the number of pathways in which each gene is involved.

**Figure 15 cells-14-00976-f015:**
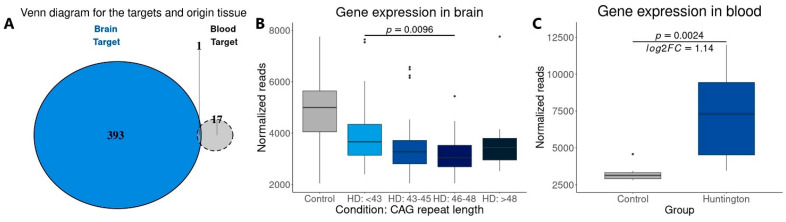
Venn diagram showing that, from the 394 HD targets identified in the prefrontal and motor cortex of HD gene-positive individuals (**A**), only FTH1 was found downregulated (in a CAG-dependent manner) (**B**) in the brain and upregulated in the blood (**C**).

**Table 1 cells-14-00976-t001:** Gene classification criteria used by BDASeq^®^.

Classification	Criteria
Zero-count genes (ZCGs)	Genes with row counts = 0
Low-count genes (LCGs)	Genes with normalized counts < 10
Equally expressed genes (EEGs)	Genes with no statistical differences between cases and controls, adjusted *p*-value (FRD ^1^) > 0.05)
Non-relevant genes (NRGs)	Genes with adjusted *p*-value (FRD) < 0.05, but |log2FC| < 0.40
Downregulated genes (DRGs)	Genes with adjusted *p*-value < 0.05 and log2FC < −0.40
Upregulated genes (URGs)	Genes with adjusted *p*-value < 0.05 and log2FC < 0.40

^1^ FDR—false discovery rate.

**Table 2 cells-14-00976-t002:** Target genes involved in HD-related pathophysiological process.

Process	Target Genes Identified	Pathophysiological Process	Reference
Metal ion response	*CDH1, MMP9, MT1, MTIF, MT1G, MTIH, MT1M, MT1X, MT2A*	- Copper, manganese, and zinc are increased in CSF in early HD - Copper and iron are increased in striatum of HD, causing pro-oxidant effects and neurodegeneration - Copper and zinc increased mHTT aggregation	Pfalzer et al. [44] Fox et al. [45] Rosas et al. [46] Cordeiro et al. [47]
Cholesterol metabolism	*CD36, DHCR24, FASN, HMGCS1, MVD, TM7SF2*	- Cholesterol is an essential membrane component in the CNS - Cholesterol cannot cross the blood–brain barrier (BBB) due to its association with lipoproteins, so it is synthesized locally in the CNS - HTT protein is a key regulator of lipid metabolism - plasma 24S-hydroxycholesterol is reduced in HD patients proportionally to disease progression - mHTT decreases the maturation of SREBP and the upregulation of LXR and LXR-targeted genes (SREBP, ABCG1, and ABCG4, HMGCoA reductase, ApoE), resulting into a lower synthesis and transport of cholesterol from astrocytes to neurons via ApoE - HD attenuates brain sterol synthesis and accumulation of cholesterol in neuronal membranes, leading to neuronal loss	Kacher et al. [48] Leoni et al. [49] Karasinka et al. [50]
Neuronflammation	*ADCY1, CAMKK2, CHRNB2, PB41L3, KCNA1, KCNC3, KCNC3 KCNJ12, NRG1, NRXN3, PPIA4, PRKKCG, SYT2*	- mHTT causes inflammatory responses in both CNS and peripheral tissues - increase in multiple cytokines in the plasma (IL-1β, IL-4, IL-6, IL-8, TNF-α, IL-10) and striatum (IL-6, IL-8 and TNF-α) of HD patients - Patients with HD have significantly lower plasma concentrations of IL-4, a marker of responses from T-helper-2 cells, than healthy control	Jia et al. [51] Björkqvist et al. [52] Silvestrini et al. [53] Du et al. [54] Ellrichmann et al. [55] Björkqvist et al. [52]
Transport across the brain-blood barrier	*ABCC4, ANXA3, BTG1, ITGB8, SFRP2, SIRT1, SLC16A12, SLC22A2, SLC22A8, SLC38A2, SLC5A3, SCL7A2, TWIST1*	- mHTT expression alters the neurovasculature by increasing cerebral blood volume, small vessel density, and BBB permeability in rodent models of HD and patient tissue	Katt et al. [56] Knox et al. [57] Hua et al. [58] Lin et al. [59] Vignone et al. [60] Pardo et al. [61] Yaun et al. [62]
Axon guidance	*AZIN2, CHRNB2, CNTN6, DCTN1, DPYSL5, EPHA10, HTR2C, KCNA1, KCNA3, MAP1B, NEFH, PARD3, PARD6B, POU4F1, PRKCG, RGS11, RIT2, ROCK1, SCN1B, SDC2, SEMA4D, SEMA5B, SGK1, SLC30A3, SLC39A12, SYT2*	- mHTT causes developmental defects affecting axon growth - mHTT causes axon growth dysfunctions	Capizzi et al. [63] Li et al. [64]
Neurotransmission	*ATP2B2, ATP2C2, ATP6V0E2, ACNA1H, CALHM1, CHRNB2, DHCR24, GPLD1, KCNA1, KCNC3, KCNS2, MECR, NRG1, OTOF, P2RX2, P2RX6, PLA2G4C, PLBD2, PNPLA3, RIT2, SCN1B, SCN4B, SEMA4D, SEMA5B, SLC30A3, SMPD3, SV2C, SYT2, TM7SF2, TMEM163, TRPM1, VAMP1*	- neurotransmission is altered in HD, mainly GABAergic neurotransmission	Garret et al. [65] Bruyin [66]

**Table 3 cells-14-00976-t003:** Genes identified by the BDASeq AI algorithm as CAG-repeat length-related targets.

Gene	Classification	*p*-Value	Log2FC	FDR	Pathways ^1^
*TRMT9B*	DRG	0.0388	−0.5504	0.0001	0
*SPINK8*	DRG	0.0480	1.1801	0.0000	1
*GCNT4*	DRG	0.0388	−0.4312	0.0004	0
*DYDC2*	DRG	0.0321	−0.8206	0.0003	0
*TNIP3*	DRG	0.0264	−1.7133	0.0000	0
*FAM83H*	URG	0.0189	−0.5040	0.0001	0
*RNF123*	URG	0.0216	−0.4283	0.0000	0
*SPON2*	URG	0.0099	−0.8406	0.0000	0
*ZSCAN1*	URG	0.0215	−0.7667	0.0000	0
*GLB1L2*	URG	0.0264	−0.5157	0.0000	1
*TMEM236*	URG	0.0123	1.5482	0.0000	0
*DHDDS*	URG	0.0263	−0.4200	0.0000	23

^1^ Number of pathways in which each gene is involved.

**Table 4 cells-14-00976-t004:** Genes identified by the BDASeq AI algorithm as onset age-related targets.

Gene	Classification	*p*-Value	Log2FC	FDR	Pathways ^1^
*OR3A2*	DRG	0.0292	−0.5151	0.0064	0
*PRCD*	DRG	0.0007	−0.4182	0.0025	0
*CD177*	DRG	0.0092	1.9313	0.0162	0
*VWA7*	DRG	0.0216	−0.7744	0.0001	0
*PDZD7*	DRG	0.0052	−0.5045	0.0000	1
*NRBP2*	DRG	0.0415	−0.4667	0.0000	0
*EPHA10*	DRG	0.0178	−0.6829	0.0000	2
*SCN4B*	DRG	0.0439	−0.5889	0.0000	22
*CIDEA*	DRG	0.0086	−0.6417	0.0004	1
*PLEKHD1*	DRG	0.0006	−0.7914	0.0001	0
*CES4A*	DRG	0.0292	−0.6246	0.0000	1
*FASN*	DRG	0.0464	−0.4322	0.0000	2
*FTH1*	DRG	0.0392	−0.5208	0.0000	0
*CORO6*	DRG	0.0099	−0.4891	0.0000	0
*MVD*	DRG	0.0027	−0.5133	0.0004	5
*MAMDC2*	DRG	0.0042	0.5831	0.0004	0
*MTHFD2L*	DRG	0.0121	0.4417	0.0002	0
*TRIM17*	DRG	0.0292	−0.4646	0.0000	0
*NRG1*	DRG	0.0121	−0.6523	0.0000	2
*CEP112*	DRG	0.0403	0.5611	0.0000	0
*SYT2*	DRG	0.0049	−1.7452	0.0000	37
*STAC2*	DRG	0.0002	−0.5169	0.0000	6
*CBLN3*	DRG	0.0049	−1.7130	0.0001	0
*VAMP1*	DRG	0.0002	−0.6175	0.0000	25
*TP53TG5*	DRG	0.0464	−0.7439	0.0000	0
*WNT3*	DRG	0.0086	0.5000	0.0001	21
*SCN1B*	DRG	0.0027	−0.5948	0.0000	56
*PLA2G4C*	DRG	0.0025	−0.4849	0.0000	3
*SEC14L5*	DRG	0.0167	−0.5010	0.0000	0
*P2RX6*	DRG	0.0086	−0.5360	0.0000	13
*MYO15A*	DRG	0.0001	−0.5266	0.0000	6
*RGS11*	DRG	0.0080	−0.4597	0.0001	1
*TNIP3*	DRG	0.0074	−1.7133	0.0000	0
*MRC1*	URG	0.0464	1.8902	0.0000	4
*STUM*	URG	0.0229	−0.8045	0.0000	0
*KCNJ12*	URG	0.0392	−0.5354	0.0001	8
*TMEM151B*	URG	0.0244	−0.4485	0.0000	0
*FAM91A1*	URG	0.0167	0.4735	0.0000	2
*TCP11L1*	URG	0.0147	−0.4481	0.0000	1
*PDGFD*	URG	0.0427	1.0627	0.0007	9
*SHOX2*	URG	0.0006	2.6702	0.0000	16
*ABCA1*	URG	0.0370	0.9330	0.0000	16
*CA10*	URG	0.0019	−0.6081	0.0000	0
*SPHKAP*	URG	0.0275	−0.9092	0.0000	0
*SRD5A1*	URG	0.0329	−0.4246	0.0000	1
*ARPP19*	URG	0.0329	−0.4803	0.0000	1
*FEZF1*	URG	0.0002	1.6361	0.0000	6
*SIX1*	URG	0.0310	1.9791	0.0058	5
*TWIST1*	URG	0.0027	1.0096	0.0009	68
*SLC9A2*	URG	0.0229	0.6608	0.0007	0
*GABRR2*	URG	0.0339	1.3858	0.0000	1
*SMPD3*	URG	0.0121	−0.4803	0.0000	6
*TBX5*	URG	0.0109	2.0175	0.0000	37
*PPEF1*	URG	0.0001	−1.1849	0.0000	0
*TMEM38A*	URG	0.0349	−0.5346	0.0000	7
*ARHGAP10*	URG	0.0010	0.6159	0.0000	4
*PITX1*	URG	0.0025	7.0377	0.0000	19
*RCN1*	URG	0.0023	0.6193	0.0000	4
*NRXN3*	URG	0.0392	−0.5461	0.0000	10

^1^ Number of pathways in which each gene is involved.

**Table 5 cells-14-00976-t005:** Genes identified by the BDASeq^®^ AI algorithm as brain degeneration-related targets.

Gene	Classification	*p*-Value	Log2FC	FDR	Pathways ^1^
*NUDT18*	DRG	0.0060	−0.7345	0.0000	0
*CLEC2L*	DRG	0.0008	−0.7127	0.0000	0
*TIMM23B*	DRG	0.0067	−0.4999	0.0013	2
*IQANK1*	DRG	0.0406	−0.7817	0.0000	0
*SMIM10L2B*	DRG	0.0309	−0.6233	0.0000	0
*KAZN*	DRG	0.0084	−0.4590	0.0000	1
*PDZD7*	DRG	0.0141	−0.5045	0.0000	1
*AHNAK2*	DRG	0.0372	−0.6515	0.0000	0
*NRBP2*	DRG	0.0256	−0.4667	0.0000	0
*EPHA10*	DRG	0.0093	−0.6829	0.0000	2
*NXPH4*	DRG	0.0339	−1.0172	0.0000	1
*SCN4B*	DRG	0.0339	−0.5889	0.0000	22
*CIDEA*	DRG	0.0042	−0.6417	0.0004	1
*PLEKHD1*	DRG	0.0083	−0.7914	0.0001	0
*PHOSPHO1*	DRG	0.0406	−0.5368	0.0008	2
*CES4A*	DRG	0.0444	−0.6246	0.0000	1
*LDLRAD4*	DRG	0.0075	−0.4421	0.0000	0
*HR*	DRG	0.0030	−0.7181	0.0000	0
*CORO6*	DRG	0.0174	−0.4891	0.0000	0
*MVD*	DRG	0.0192	−0.5133	0.0004	5
*NRG1*	DRG	0.0142	−0.6523	0.0000	2
*ZSCAN1*	DRG	0.0093	−0.7667	0.0000	0
*RIT2*	DRG	0.0406	−0.7547	0.0000	3
*TMEM163*	DRG	0.0075	−0.5153	0.0001	6
*PLBD2*	DRG	0.0157	−0.4840	0.0000	0
*PCYOX1L*	DRG	0.0079	−0.6316	0.0001	0
*SYT2*	DRG	0.0309	−1.7452	0.0000	37
*STAC2*	DRG	0.0014	−0.5169	0.0000	6
*VAMP1*	DRG	0.0030	−0.6175	0.0000	25
*KIF12*	DRG	0.0484	−0.8271	0.0000	4
*ANGPTL2*	DRG	0.0233	−0.8151	0.0004	1
*ACP2*	DRG	0.0192	−0.5411	0.0000	1
*KCNC3*	DRG	0.0256	−0.5356	0.0000	30
*ARHGEF2*	DRG	0.0093	−0.4720	0.0000	8
*MECR*	DRG	0.0038	−0.4410	0.0000	12
*SLC30A3*	DRG	0.0484	−0.7125	0.0000	6
*VIPR1*	DRG	0.0115	−0.4111	0.0001	1
*SCN1B*	DRG	0.0002	−0.5948	0.0000	56
*NAPA*	DRG	0.0030	−0.4614	0.0000	11
*NEFM*	DRG	0.0128	−0.6006	0.0000	2
*SEC14L5*	DRG	0.0038	−0.5010	0.0000	0
*NUP93*	DRG	0.0128	−0.4756	0.0000	1
*GSTZ1*	DRG	0.0406	−0.5657	0.0001	1
*P2RX6*	DRG	0.0173	−0.5360	0.0000	13
*MYO15A*	DRG	0.0406	−0.5266	0.0000	6
*ADAM11*	DRG	0.0354	−0.5504	0.0000	4
*F7*	DRG	0.0060	−0.8341	0.0000	1
*SLC5A3*	URG	0.0164	0.8459	0.0000	4
*HMGN5*	URG	0.0256	0.5588	0.0002	2
*TOPORS*	URG	0.0067	0.4285	0.0001	13
*PTAR1*	URG	0.0211	0.6316	0.0000	0
*PLEKHF2*	URG	0.0115	0.6301	0.0000	0
*GPR171*	URG	0.0067	1.8061	0.0000	2
*SLFN11*	URG	0.0406	1.0408	0.0000	7
*BCL2*	URG	0.0060	0.5718	0.0000	95
*ARHGAP42*	URG	0.0142	0.7404	0.0000	7
*STOX1*	URG	0.0211	0.5138	0.0000	14
*RHOBTB3*	URG	0.0192	0.7695	0.0000	2
*RANBP3L*	URG	0.0157	1.2461	0.0000	5
*PPP4R2*	URG	0.0309	0.4286	0.0000	4
*GRAMD2B*	URG	0.0060	0.8249	0.0000	0
*GABPA*	URG	0.0060	0.4045	0.0002	7
*BCL2L11*	URG	0.0339	0.5653	0.0002	35
*GXYLT1*	URG	0.0053	0.4231	0.0002	1
*PARD3*	URG	0.0339	0.9958	0.0000	8
*SLC39A12*	URG	0.0233	1.1799	0.0000	5
*G3BP1*	URG	0.0034	0.4533	0.0000	8
*USP53*	URG	0.0406	0.8390	0.0000	2
*YAP1*	URG	0.0115	1.1163	0.0000	57
*SLC38A2*	URG	0.0173	0.8775	0.0000	8
*MAP7D3*	URG	0.0030	0.5214	0.0019	0
*FEZF1*	URG	0.0309	1.6361	0.0000	6
*GNA13*	URG	0.0104	0.7054	0.0000	15
*UPK1B*	URG	0.0173	2.6019	0.0000	1
*GMNN*	URG	0.0339	0.7335	0.0001	9
*GABRR2*	URG	0.0174	1.3858	0.0000	1
*DSE*	URG	0.0084	0.9154	0.0000	14
*ITGB8*	URG	0.0021	0.6760	0.0000	20
*ATP11C*	URG	0.0371	0.6688	0.0000	1
*EZR*	URG	0.0406	0.7495	0.0000	20
*TGFBR3*	URG	0.0211	1.1844	0.0000	34
*LAPTM4A*	URG	0.0443	0.5393	0.0000	0
*IL17RB*	URG	0.0173	1.1067	0.0000	5
*RCN1*	URG	0.0142	0.6193	0.0000	4

^1^ Number of pathways in which each gene is involved.

**Table 6 cells-14-00976-t006:** Genes identified by the BDASeq AI algorithm as lifespan-related targets.

Gene	Classification	*p*-Value	Log2FC	FDR	Pathways ^1^
*MRC1*	DRG	0.0412	1.8902	0.0000	4
*STUM*	DRG	0.0090	−0.8045	0.0000	0
*GAS1*	DRG	0.0152	1.0313	0.0001	11
*NPAS4*	DRG	0.0075	−2.8925	0.0049	0
*SPTLC3*	DRG	0.0075	1.9498	0.0001	0
*S100P*	DRG	0.0427	2.4416	0.0000	5
*SLC16A12*	DRG	0.0112	3.0162	0.0000	2
*OSR1*	DRG	0.0028	1.3507	0.0036	8
*STRA6*	DRG	0.0026	1.3572	0.0138	2
*CEMIP*	DRG	0.0152	0.7861	0.0000	2
*FCGR2B*	DRG	0.0023	2.3809	0.0000	53
*PKP2*	DRG	0.0230	1.0042	0.0001	9
*MARCO*	DRG	0.0128	2.1951	0.0000	3
*TMEM163*	URG	0.0341	−0.5153	0.0001	6
*ANGPT2*	URG	0.0165	1.4528	0.0000	17
*RIPOR3*	URG	0.0427	1.2127	0.0000	0

^1^ Number of pathways in which each gene is involved.

**Table 7 cells-14-00976-t007:** Genes identified as DEGs in the blood of HD gene-positive individuals.

Gene ID	Gene	Log2FC	FDR	Classification
ENSG00000250568 ^1^	*PABPC1*	1.2687	0.0267	URG
ENSG00000260035	*Lnc-DUOXA1-2*	2.3900	0.0003	URG
ENSG00000149925	*ALDOA*	0.8515	0.0014	URG
ENSG00000141480	*ARRB2*	0.4655	0.0009	URG
ENSG00000135636	*DYSF*	0.6159	0.0109	URG
ENSG00000132589	*FLOT2*	0.4394	0.0369	URG
ENSG00000167996	*FTH1*	1.1435	0.0024	URG
ENSG00000174021	*GNG5*	0.4621	0.0295	URG
ENSG00000170837	*GPR27*	0.6579	0.0127	URG
ENSG00000030582	*GRN*	0.5643	0.0495	URG
ENSG00000248099	*INSL3*	1.4028	0.0014	URG
ENSG00000251230	*MIR3945HG*	0.8796	0.0493	URG
ENSG00000170909	*OSCAR*	0.5463	0.0403	URG
ENSG00000170525	*PFKFB3*	0.8153	0.0006	URG
ENSG00000105287	*PRKD2*	0.4599	0.0077	URG
ENSG00000112053	*SLC26A8*	0.9719	0.0495	URG
ENSG00000266402	*SNHG25*	1.6604	0.0156	URG
ENSG00000184292	*TACSTD2*	3.8049	0.0000	URG
ENSG00000156414	*TDRD9*	1.1963	0.0403	URG
ENSG00000170873	*MTSS1*	−0.6198	0.0011	DRG

^1^ Pseudogene.

## Data Availability

RNA sequencing files analyzed in this study are available on the SRA database (https://www.ncbi.nlm.nih.gov/sra, accessed on 15 June 2024).

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
