# Peer review of "From Brain to Blood: Uncovering Potential Therapeutical Targets and Biomarkers for Huntington’s Disease Using an Integrative RNA-Seq Analytical Platform (BDASeq®)"

_cells, 2025, doi:10.3390/cells14130976_

Round 1

Reviewer 1 Report

Comments and Suggestions for Authors

The authors perform the differential expression analysis along with machine learning algorithms were coupled to identify differentially expressed genes and potential biomarkers in huntington's disease. They identified 5,834 DEGs, including 394 putative druggable genes involved in processes like neuroinflammation, metal ion dysregulation, and blood-brain barrier dysfunction. They proposed FTH1 as a promising biomarker for HD, with its expression down regulated in the prefrontal cortex and upregulated in peripheral blood in a CAG repeat-dependent manner. The approach used by authors is scientifically sound. However, I have a few suggestions regarding specific areas that need addressing to maximize the manuscript’s suitability for publication.

Comments

[1] In title From Brain to Blood: Uncovering Potential Therapeutical Targets and Biomarkers for Huntington’s Disease Using an Intregrative RNA-Seq Analytical Plataform(BDASeq®)" Please confirm “Plataformor “Platform”.

[2] At Page 7 and Line no. 215. The phrase “Specifically, Therefore, exclusion included:” is redundant and awkward. Consider revising to something like: “Specifically, the following criteria were used for exclusion”.

[3] In the sentence "Consequently, 61 unmatched control samples (80.8 ± 14.4 years,) were excluded", the comma after the parenthesis should be removed.

[4] Highlighting that this is “the largest transcriptomic study of HD performed to date” is impactful. Consider supporting this statement with a citation or comparison to prior studies to substantiate the claim.

[5] At Page no. 8 and Line no. 254. The removal of 20 outlier samples is an important quality control step. It would be helpful to provide more detail on the criteria used for their exclusion—e.g., were they identified via distance to cluster centroids, silhouette scores, or manual inspection?

[6] “transcriptomic profiler” should be corrected to “transcriptomic profiles.”

[7] Line 243 ends with a “t” that should be removed.

[8] The sentence starting with “Coupling both unsupervised and clustering techniques...” could be revised to avoid redundancy, as clustering is an unsupervised technique. Consider rephrasing for clarity.

[9] Page no. 8 and Line no. 265. lacking a meaningful log2FC” in defining NRGs. Please define what threshold constitutes "meaningful" (e.g., |log2FC| ≥ 1 or so ) and ensure it's consistently applied throughout the analysis.

[10] Page no.11 and Section 3.4. The enrichment results are summarized broadly (e.g., "responses to metal ions"), but it's unclear how many genes contributed to each term and the specific enrichment statistics (e.g., p-value, fold enrichment). A more complete presentation either in-text or in supplementary table would enhance transparency and reproducibility.

[11] Page no.11 and Section 3.4. The manuscript could benefit from discussing how these aging-associated genes interact with known HD molecular mechanisms. Are these genes also differentially expressed in aging brains without HD, or are they uniquely altered in HD, suggesting disease-specific acceleration of aging processes?

[12] Page no. 23. It is stated that CAG repeat expansion does not regulate them directly, yet their expression reflects disease state. Consider including evidence or discussion about the mechanistic pathways (e.g., stress signaling, metabolic shift) connecting mHTT pathology with their regulation.

[13] The exclusion of two non-coding DEGs from the blood analysis is mentioned, but the justification is not entirely clear. Please elaborate. Were these genes excluded purely due to their non-coding status, or due to lack of known function or poor expression specificity? Given the emerging role of non-coding RNAs in neurodegenerative diseases, this decision should be justified more robustly.

[14] At Page no 24. making of this largest transcriptomic study” → corrected to “making this the largest transcriptomic study.

[15] Figure 13. Tha gene lables were overlapped. It is hard to visualize. Authors should provide the figure with better resolution.

[16] In table 7. The Gene name of “ENSG00000250568”  and “ENSG00000260035” is missing.

Addressing these points will likely enhance the manuscript's quality and increase its chances of acceptance.

Comments on the Quality of English Language

No comments

Author Response

Dear Reviewer,

Thank you for your comments. To facilitate the review process, we answer each question.

[1] In title From Brain to Blood: Uncovering Potential Therapeutical Targets and Biomarkers for Huntington’s Disease Using an Intregrative RNA-Seq Analytical Plataform(BDASeq®)" Please confirm “Plataform” or “Platform”.

We apologize for this typo, the word “plataform” was replaced to platform

 [2] At Page 7 and Line no. 215. The phrase “Specifically, Therefore, exclusion included:” is redundant and awkward. Consider revising to something like: “Specifically, the following criteria were used for exclusion”.

We apologize for the sentence. In fact, there was a mistake in this sentence. As suggested, the sentence was rewritten to “Specifically, the following criteria were used for exclusion”.

[3] In the sentence "Consequently, 61 unmatched control samples (80.8 ± 14.4 years,) were excluded", the comma after the parenthesis should be removed.

As observed, the comma was inappropriately inserted. As requested, the comma was excluded.

[4] Highlighting that this is “the largest transcriptomic study of HD performed to date” is impactful. Consider supporting this statement with a citation or comparison to prior studies to substantiate the claim.

You agree that this sentence is impactful. However, in fact, this was the largest transcriptomic study of HD already done, as recognized by the International Huntington Association (IHA), in the two last editions of the MENA Congress for Rare Disease. Moreover, this recognition was also obtained through FAPESP, who was responsible for the financial support of this study (https://agencia.fapesp.br/a-startup-supported-by-fapesp-is-conducting-one-of-the-largest-studies-on-huntingtons-disease/51647). These references were added to the sentence as requested.

[5] At Page no. 8 and Line no. 254. The removal of 20 outlier samples is an important quality control step. It would be helpful to provide more detail on the criteria used for their exclusion—e.g., were they identified via distance to cluster centroids, silhouette scores, or manual inspection?

We apologize for the short description of this important step. In fact, to remove the outlier’s samples we used the combination of 2D UMAP representation of the bulk gene expression – low count and zero count genes were removed - and combine with DBSCAN analysis. The best parameter for fine tuning were found to be the best combination which could prove the highest silhouette score. DBSCAN in nature works by defining clusters as densely packed regions separated by regions of lower density. This approach allows DBSCAN to discover and label sparse distributed samples as noise, i.e. outliers. These details were added from line 261 to clarify the criteria used for the outlier exclusion.

[6] “transcriptomic profiler” should be corrected to “transcriptomic profiles.”

We apologize for this typo. The word “profiler” was replaced by profiles

[7] Line 243 ends with a “t” that should be removed.

We apologize for this mistake. The “t” was removed

[8] The sentence starting with “Coupling both unsupervised and clustering techniques...” could be revised to avoid redundancy, as clustering is an unsupervised technique. Consider rephrasing for clarity.

The sentence was corrected to “Using clustering techniques…”

[9] Page no. 8 and Line no. 265. lacking a meaningful log2FC” in defining NRGs. Please define what threshold constitutes "meaningful" (e.g., |log2FC| ≥ 1 or so ) and ensure it's consistently applied throughout the analysis.

The criteria for NRGs definition are shown in Table 1.

[10] Page no.11 and Section 3.4. The enrichment results are summarized broadly (e.g., "responses to metal ions"), but it's unclear how many genes contributed to each term and the specific enrichment statistics (e.g., p-value, fold enrichment). A more complete presentation either in-text or in supplementary table would enhance transparency and reproducibility.

Please, verify Excel Supplementary S4. In this file, we show all details about the enrichment results, describing the pathway name, the number of genes overlap, p-value and FRD, Status of the pathways (up- or downregulated), the complete list of genes belonging to the pathway that were identified in this study, Gene ID, Log2FC, Gene classification (URG or DRG) and their respective class. This supplementary material was added to enhance transparency and reproducibility, as suggested.

[11] Page no.11 and Section 3.4. The manuscript could benefit from discussing how these aging-associated genes interact with known HD molecular mechanisms. Are these genes also differentially expressed in aging brains without HD, or are they uniquely altered in HD, suggesting disease-specific acceleration of aging processes?

In section 3.4 we describe that the putative druggable genes identified by BDASeq® is involved in biological process and molecular functions related to the HD pathophysiology. In last year, we conducted an independent study discussing how the aging is related to HD, and how the use of controls older than cases (HD-positive individuals) can contribute to both false-positive and false-negative results. For this discussion, please, verify the publication Dias-Pinto et al. (2024), available on https://www.frontiersin.org/journals/genetics/articles/10.3389/fgene.2024.1377237/full

[12] Page no. 23. It is stated that CAG repeat expansion does not regulate them directly, yet their expression reflects disease state. Consider including evidence or discussion about the mechanistic pathways (e.g., stress signaling, metabolic shift) connecting mHTT pathology with their regulation.

Thank you for this comment. It is recognized that the number of CAG repeat on PolyQ tract of HTT gene is closely related to the onset age. In our study, we observed that FTH1, the only gene that was identified as commonly deregulated in brain and blood, has an expression level reduced in individuals with more than 43 GAC repeats, than those with 36-42 CAG repeats, as demonstrated in Figure 15. However, establishing a possible connection between FTH1 with a specific pathway could induce a false-positive result, attributing this effect to a set of genes. For this reason, we understand that further studies are required to provide this correlation, as we included on page 23.

[13] The exclusion of two non-coding DEGs from the blood analysis is mentioned, but the justification is not entirely clear. Please elaborate. Were these genes excluded purely due to their non-coding status, or due to lack of known function or poor expression specificity? Given the emerging role of non-coding RNAs in neurodegenerative diseases, this decision should be justified more robustly.

We apologize if the justification was not entirely clear. To clarify this point, we informed that the focus of this study was concentrated on identifying protein-conding genes (mRNAS) that can serve as potential druggable target or biomarkers, since these genes are most suitable for drug design and/or biochemical analysis using antibodies. However, the two non-coding DEGs are described in Table 7 (PABPC1 and Lnc-DUOXA1-2).

[14] At Page no 24. making of this largest transcriptomic study” → corrected to “making this the largest transcriptomic study.

We apologize for the mistake in the sentence. It was corrected as requested.

[15] Figure 13. Tha gene lables were overlapped. It is hard to visualize. Authors should provide the figure with better resolution.

Unfortunately, this overlapping is expected for this type of visualization, since the sizes are represented according to the coordinated based on the Gene importance (Y axis) and log2FC (X axis). However, details about the genes are clearly shown in Table 5.

[16] In table 7. The Gene name of “ENSG00000250568” and “ENSG00000260035” is missing

The gene names were added in Table 7: ENSG00000250568 (pseudogene PABPC1) and ENSG00000260035 (Lnc-DUOXA1-2)

Reviewer 2 Report

Comments and Suggestions for Authors

This is a very interesting and well-written paper, both in the design, the results (that are relevant), and the discussion. I suggest that the authors should discuss the strengths and limitations of the study, and to suggest drugs that could be potentially used in the treatment of Huntington's disease according to the results obtained.  

Author Response

This is a very interesting and well-written paper, both in the design, the results (that are relevant), and the discussion. I suggest that the authors should discuss the strengths and limitations of the study, and to suggest drugs that could be potentially used in the treatment of Huntington's disease according to the results obtained.

Thank you for your comments and suggestions. The major limitation is the need of further validation of the putative target genes identified in this study, as already discussed on the conclusion “Despite the promising results, further validation using techniques like qRT-PCR and immunoassays is needed to establish FTH1's clinical utility as a biomarker for HD”. In this regard, we would like to inform you that a novel study, recruiting patients with HD, is ongoing to validate our results. However, this validation will be published in the next paper.  In relation to candidate drugs, we also inform that we conduct a complex analysis based on pharmacogenomics, identifying several potential drugs that could be repurposed for the HD treatment, including valproic acid. Due to the complexity of these new results, and to guarantee the transparency and reproducibility of the methods, results obtained will be published in other paper, that is under review by the authors to be submitted to Cells in next week.

Reviewer 3 Report

Comments and Suggestions for Authors

This study offers the most comprehensive RNA-Seq reanalysis of Huntington’s Disease (HD) so far, combining 353 brain cortex samples from multiple datasets using the BDASeq platform. By using propensity score matching to control for age-related factors and applying eight different differential expression methods through a recursive algorithm, the researchers identified 5,834 differentially expressed genes, including 394 potential protein-coding drug targets involved in HD. These genes relate to important processes like neuroinflammation, metal ion response, cholesterol metabolism, blood-brain barrier issues, and neuronal signaling. One standout finding is FTH1, which shows opposite expression patterns in the brain and blood depending on CAG repeat length. The study also found strong links between gene expression and clinical features like CAG repeats, age of onset, brain degeneration, and lifespan, underscoring their therapeutic potential. As the largest transcriptomic reanalysis of HD to date, it clearly shows that combining multiple analysis methods with AI, as BDASeq does, can greatly improve how efficiently and accurately targets and biomarkers are identified. Because of this, I have a very positive view of the work. The strengths of the study lie in its thorough bioinformatics approach, integrating many datasets, carefully controlling for confounders, and using AI to prioritize targets beyond just fold-change statistics. This multi-cohort RNA-Seq analysis offers valuable insights into HD’s molecular mechanisms and opens new paths for finding biomarkers and drug targets. That said, it only looks at the prefrontal and motor cortices, leaving out key regions like the striatum and caudate nucleus, which are central to HD neurodegeneration. This limitation means some important region-specific changes may be missed, which could affect how applicable the results are to the core disease process. A few additional points for the authors to consider:

  1. The study draws heavily from RNA-Seq data across 12 BioProjects in the NCBI SRA database, but it doesn’t clarify how diverse these samples are in terms of ethnicity, sex, or disease stage like early-onset HD. Without this info, there’s a risk that population biases (e.g., overrepresentation of European ancestry) could limit how well the findings apply to different groups.
  2. While the authors excluded 61 elderly control samples (average age ~81) to match ages better, they don’t discuss if this might remove important aging-related signals tied to neurodegeneration. Also, even after matching, the average age at death for HD cases (~58) is still quite a bit lower than controls (~81), and the paper doesn’t explore if HD-related health issues might influence this difference, which could confound gene expression results.
  3. Focusing only on the prefrontal and motor cortices means the study misses other key brain regions affected in HD, especially the striatum. This could overlook critical, region-specific disease mechanisms.
  4. Finally, all findings are based on bioinformatics analyses without experimental follow-up, like qPCR or protein-level validation of targets such as FTH1. This limits confidence since sequencing data can have technical biases. It would strengthen the study if the authors included or at least discussed the need for experimental validation to back up their proposed biomarkers and targets.

Author Response

The study draws heavily from RNA-Seq data across 12 BioProjects in the NCBI SRA database, but it doesn’t clarify how diverse these samples are in terms of ethnicity, sex, or disease stage like early-onset HD. Without this info, there’s a risk that population biases (e.g., overrepresentation of European ancestry) could limit how well the findings apply to different groups.

Dear Reviewer,

Thank you for this important comment. We would like to inform you that all available data of both HD-positive individuals and neurologically normal individuals (controls) were shown in Supplementary Excel S1. The file previously sent was replaced with a complete version providing all information available on the metadata of the samples, which can be also consulted in the SRA database.

While the authors excluded 61 elderly control samples (average age ~81) to match ages better, they don’t discuss if this might remove important aging-related signals tied to neurodegeneration. Also, even after matching, the average age at death for HD cases (~58) is still quite a bit lower than controls (~81), and the paper doesn’t explore if HD-related health issues might influence this difference, which could confound gene expression results.

Thank you for this comment. The influence of aging, particularly related to the use of controls with no age matching with the cases, was previously analyzed and extensively discussed by us in Dias-Pinto et al. (2024). The reference to this work is described in discussion “Reanalyzing the transcriptomic dataset of one of these studies [84], we demonstrated that failure to match cases and controls by age negatively affects DEG identification, leading to an increase in both false positive and false-negative errors [19]. Using PSM algorithm to create an age-matched control group, we were able to identify multiple putative druggable genes involved in HD pathophysiology that do not belong to heat shock family [19]”. For more details about the impacts of use of controls with no age matching with cases, please, verify our previous publications that statistically analyze this theme (link for Dias-Pinto et al. (2024): https://www.frontiersin.org/journals/genetics/articles/10.3389/fgene.2024.1377237/full)

Focusing only on the prefrontal and motor cortices means the study misses other key brain regions affected in HD, especially the striatum. This could overlook critical, region-specific disease mechanisms.

Thank you for this important comment. In fact, striatum is the most affected area, since the medium spiny neurons are most susceptible to the neurotoxic effects of mutated huntingtin. However, there is an important limitation to studying human striatum at a transcriptomic level. This is because, nearly 90% of striatal neurons are highly degenerated at the moment of death of patients with HD. However, studies based on MRI showed that, in late-stage HD, prefrontal cortex exhibits loss of projection neurons in layers III, Y, and VI and glial density increase in deeper layer (VI) consistent with cortical degeneration. Based on these evidence, transcriptomic studies have focused on prefrontal cortex. To clarify this important question, these information were added from line 528.

Finally, all findings are based on bioinformatics analyses without experimental follow-up, like qPCR or protein-level validation of targets such as FTH1. This limits confidence since sequencing data can have technical biases. It would strengthen the study if the authors included or at least discussed the need for experimental validation to back up their proposed biomarkers and targets.

Thank you for this important comment. In fact, the validation of target genes identified through RNA-Seq by qRT-PCR or immunoassays is crucial to confirm the results obtained through RNA-Seq. This study reanalyzed public datasets of RNA-Seq using BDASeq® to identify candidates to druggable targets and/or biomarkers, identifying FTH1 as a putative biomarker for Huntington’s disease. To validate the FTH1 application as a HD biomarker, we would like to inform you that now, we are conducting a novel study in Brazil, with the support of Brazil Huntington Association (ABH), that will recruit volunteers (in early-, medium- and late-stage) to analyze the FTH1 expression in blood though qRT-PCR and ELISA. The results may be published in one year in a new paper.